# Evaluation of Cloud 3D Printing Order Task Execution Based on the AHP-TOPSIS Optimal Set Algorithm and the Baldwin Effect

**DOI:** 10.3390/mi12070801

**Published:** 2021-07-06

**Authors:** Chenglei Zhang, Cunshan Zhang, Jiaojiao Zhuang, Hu Han, Bo Yuan, Jiajia Liu, Kang Yang, Shenle Zhuang, Ronglan Li

**Affiliations:** 1School of Electrical and Electronic Engineering, Shandong University of Technology, Zibo 255000, China; zcs@sdut.edu.cn; 2School of Mechanical & Vehicle Engineering, Linyi University, Linyi 276000, China; zhuangjiaojiao@lyu.edu.cn (J.Z.); hanhu@lyu.edu.cn (H.H.); 3Shandong Longli Electronic Co., Ltd., Linyi 276000, China; zhuangshenlele@163.com (S.Z.); 290852860@gmail.com (R.L.); 4School of Mechanical and Electrical Engineering, Wuhan City Polytechnic, Wuhan 430064, China; yuanbohg@126.com; 5College of Mechanical Engineering, Anyang Institute of Technology, Anyang 455000, China; angongyangkang@163.com

**Keywords:** cloud manufacturing (CMfg), 3D printing device resources, HPSO, multi-objective optimization, baldwin effect

## Abstract

Focusing on service control factors, rapid changes in manufacturing environments, the difficulty of resource allocation evaluation, resource optimization for 3D printing services (3DPSs) in cloud manufacturing environments, and so on, an indicator evaluation framework is proposed for the cloud 3D printing (C3DP) order task execution process based on a Pareto optimal set algorithm that is optimized and evaluated for remotely distributed 3D printing equipment resources. Combined with the multi-objective method of data normalization, an optimization model for C3DP order execution based on the Pareto optimal set algorithm is constructed with these agents’ dynamic autonomy and distributed processing. This model can perform functions such as automatic matching and optimization of candidate services, and it is dynamic and reliable in the C3DP order task execution process based on the Pareto optimal set algorithm. Finally, a case study is designed to test the applicability and effectiveness of the C3DP order task execution process based on the analytic hierarchy process and technique for order of preference by similarity to ideal solution (AHP-TOPSIS) optimal set algorithm and the Baldwin effect.

## 1. Introduction

Intelligent algorithms are the most commonly used tool to solve NP-complete combination optimization problems. After years of development, many different random search strategies have emerged [1]. They all form their own iterative search mechanisms by simulating the behavior and evolution of natural ecology. They are simple, universal, robust, and easy to use in parallel processing [2]. To solve the problem of computing resource allocation in a cloud 3D printing service (C3DPS) resource pool, it is necessary to consider both the constraints of the task graph and the nodes. In the neighborhood search algorithm, the typical simulated annealing and tabu search algorithms are used due to the strong randomness and the need for only a single iteration based on an individual search. There is a very low probability of finding an optimal solution within the feasible solution set during a short iteration period. Presently, this algorithm is not suitable for solving combination optimization problems such as scheduling when combined with another algorithm application [3]. However, only the heredity and immunity algorithms show good performance in various applications and other improvements in evolutionary learning algorithms. Others, such as artificial neural network and DNA calculation algorithms, are less mature in the computational process of solving problems in mechanics as applied to combination optimization. In addition, a hybrid of ant colony and particle swarm optimization (PSO) algorithms was proposed to solve the multi-objective flexible scheduling problems based on the analysis of objectives and their relationship. This self-learning algorithm describes the application of the particle swarm optimization (PSO) metaheuristic to the continuous version of the p-median problem. To a certain extent, most of the improved algorithms convert the original numerical changes into individuals in a particle swarm, and the optimum crosscurrent operation or sub-optimal exchange has no obvious practical application. The optimum crosscurrent operation is also called genetic PSO [4].

The evaluation of C3DPS quality belongs to the field of multiple-criteria decision-making (MCDM) [5]. A study of relevant data around the world reveals that research on service quality is primarily based on scientificity, reliability, a combination of quantitative and qualitative aspects, and other principles. Therefore, there are three primary methods for comprehensively evaluating C3DPS quality: the classical comprehensive evaluation method of cloud services (CSs), the CS evaluation method based on multi-attribute decision-making, and the quality of service (QoS) [6,7]. These methods are used in the analytic hierarchy process (AHP), fuzzy mathematics, the effectiveness function, principal component analysis (PCA), and so on [8]. Among these methods, analytic hierarchy process (AHP) is a combination of qualitative and quantitative decision analysis methods, and it has a wide range of practicality for decision analysis of various types of problems. For the AHP decision-making problem of multi-expert evaluation, the judgments given by experts may be very different due to the differences in cognition of different experts, the differences in preferences, and the ability of experts. Fuzzy mathematics is an extension or a super-set of the Boolean logic, aimed at maintaining the concept of the partial truth. The utility function method is usually employed to represent the relationship between the utility obtained by the consumer in consumption and the quantity of the product portfolio [9]. The PCA is a statistical method. The main concept behind the PCA is to consider the correlation among features. If the correlation is very high among a subset of the features, PCA will attempt to combine the highly correlated features and represent this data with a smaller number of linearly uncorrelated features. The value of the PDF at any given point in the set of possible values taken by the random variable can be interpreted as providing the relative likelihood that the value of a random variable will equal to the sample. The TOPSIS method sorts potential solutions according to the closeness among a limited number of negative ideal solutions and the preference by similarity to ideal solution (PSIS), which is a comparison of the relative advantages and disadvantages of existing objects [10]. The DEA method is a quantitative analysis method that can input and output several indicators. It measures the linear programming efficiency of the amendment of a decision-making unit, by using data envelopment analysis (DEA) [11].

This study is conducted with the aims to develop a new ranking method for multicriteria decision-making problems with conflicting criteria. These methods include the AHP method, fuzzy mathematics, the effectiveness function, the PCA method, and the PDF method [12]. It performs a literature review of common multicriteria decision making methods, examines the advantages and disadvantages of the identified methods, and explains how their common applications relate to their relative strengths and weaknesses. The TOPSIS method is suitable for small restrictions on the number and types of evaluation indicators. It can be objectively optimized for several decision-making options [13]. The need for optimization of multiple objectives to reach a set of optimal solutions has recently been increased in many applications. Therefore, various types of multi-objective optimization algorithms have been developed for solving multi-objective problems. The Pareto optimal set algorithm has the advantages of simplicity, good convergence, and fast search speed. This algorithm retains the advantages and addresses the disadvantages of the improved local optimal solution or global optimal solution algorithm; its use in solving the resource allocation problem is relatively suitable and has potential.

The research on index systems is mainly divided into the reputation evaluation index, QoS evaluation index, service resource combination evaluation index, and other evaluation indexes [14]. According to the characteristics of the evaluation index, it is a cloud service evaluation method for multi-attribute decision-making. This method puts fewer restrictions on the characteristics of the evaluation index and the type of index and can optimize multiple decision-making schemes. An intelligent optimization algorithm has a simple algorithmic principle. The convergence and search speed are good, the evaluation efficiency is high, it is relatively easy to obtain satisfactory evaluation results, and the related research fields are expanding. Therefore, this paper proposes a multi-attribute decision service evaluation method based on an intelligent optimization algorithm, that is, a multi-attribute decision evaluation method based on a Pareto optimal set algorithm. To solve the problem of computing resource allocation, it is necessary to consider the constraints of both the task graph edges and nodes in the solution process of the C3DPS resource pool [15]. Among neighborhood search algorithms, typical simulated annealing and abut search have strong randomness and use only a single individual based on the iterative search, so the probability of finding a feasible solution is extremely low in a short time (iteration time). At present, they are mostly combined with other algorithms, but these are not suitable for solving optimization problems such as scheduling combinations.

Among evolutionary learning algorithms, only immune algorithms and genetic algorithms have been proven to show good performance in various applications and improvements. Other algorithms are less developed or more immature in the neighborhood of optimization, such as artificial neural networks and DNA computing. In addition, the ant colony algorithm and particle swarm algorithm are widely used in scheduling problems. However, the self-learning mechanism of PSO was originally designed for continuous numerical optimization problems [16]. For discrete combination optimization problems, the speed and position learning mechanisms of these algorithms are not suitable. Most of the improvements convert the original numerical change into a crossover operation or exchange sub-operation between an individual and the current optimal solution. To a certain extent, this method is transformed into a genetic particle swarm algorithm, and the effect is not obvious in practical applications. In summary, the intelligent optimization algorithm for a Pareto optimal set is based on the above-mentioned advantages and disadvantages [17,18,19]. It is a newly improved local optimal solution or global optimal solution algorithm that is relatively suitable and has great potential for solving the problem of computing resource allocation.

## 2. Framework for Cloud Service Evaluation Based on a Hybrid Multi-Objective BM-MOPSO Evaluation Model

Pareto optimality is a cornerstone concept in the field of optimization. Based on Pareto optimality theory and the AHP-TOPSIS evaluation model, this framework is objectively determined to the weight of each evaluation set and indicator of the AHP method MATLAB software. The TOPSIS method is analyzed to the closeness of the optimal solution and determined by the overall similarity between an optimal and an ideal solution [20]. Therefore, it is possible to construct a new AHP-TOPSIS evaluation ideal value approximation model for decision-making, which can be used in decision-making.

This cloud service evaluation is as follows: The task-order requirements are normalized by task decomposition and detailed to task-order requirements in this C3DPS platform. This task-order requirement uses a parsing function and an analysis task that forms a one-to-one atomic task [21]. According to the atomic task, search matching is used to perform supply-and-demand matching of service resources, form a dual feasible solution for the C3DPS resource candidate service set, and provide feedback to users. Therefore, Pareto optimal optimization algorithm is objectively determined to the weight of each evaluation of set and indicator and construct a comprehensive AHP-TOPSIS evaluation model based on a Pareto optimal set. Here, Pareto optimal optimization algorithm and many local search strategies perform within the multi-objective Evaluation method. This framework of the evaluation method of the C3DPS order task execution process based on the AHP-TOPSIS optimal set algorithm and Baldwin effect is shown in Figure 1.

Here, task-order requirement agents: each agent is registered and released to all the requirements as the condition attribute and forms a meta-model corresponding to the task-order demand; task-order decomposition agent: According to the task decomposition strategy of the C3DPD Platform, this meta-model $OR_{model}$ is expressed by the discrete attributes values of the decision table; Search-matching agent: this agent is a domain $ORS=(ORS_{1},ORS_{2},\ldots ,ORS_{n})$ of feasible solutions of the corresponding service resource candidate service subset [22]. Therefore, $ORS_{i}$ is a subset of candidate services, and these are matched by the atomic task $OR_{i}$; Service resource agent: according to the weight indicators of the domain $ORS=(ORS_{1},ORS_{2},\ldots ,ORS_{n})$ of feasible solutions, it corresponds to some subsets of candidate services by the Pareto optimal optimization algorithm, also supported to this C3DPS platform; Evaluation agent: the domain $ORS=(ORS_{1},ORS_{2},\ldots ,ORS_{n})$ of feasible solutions are solved that it corresponds to some subsets of this candidate services.

## 3. Intelligent Optimization Algorithm for the Pareto Optimal Set and AHP

### 3.1. Intelligent Optimization Algorithm for Pareto Optima

The Pareto optimal condition is as follows: In the process of information resource allocation, the marginal rate of technical substitution of two kinds of information resource *X* and *Y* in any computer resource is equal to the marginal rate of technical substitution of producing these two kinds of information resources *X* and *Y*; that is,
(1)$${MRT}_{XY}=MRS_{XY}^{A}$$
where MRT is an ideal state of a computing resource allocation and $MRS_{XY}^{A}$ is a marginal rate of technical substitution of the two kinds of information resource *X* and *Y* in any computing resource.

The intelligent optimization algorithm for Pareto optima is derived from the above concept [23]. Based on the analysis of the existing intelligent optimization search strategies, a multi-objective optimization algorithm for Pareto optimal and AHP-TOPSIS evaluation models is proposed. It is a multi-objective evolutionary algorithm that combines evolutionary computation and a multi-local search strategy.

**Definition** **1.**
*Assuming that the outer set is in the i-th generation, each individual is assigned an intensity value, namely,*
(2)$$S_{i}^{\left(k\right)}=\frac{t_{i}^{\left(k\right)}}{n^{\left(k\right)}+1}\begin{array}{cc} & i=1,2...,n_{T}^{\left(k\right)} \end{array},$$
*where*
$t_{i}^{\left(k\right)}=|\{(u|u_{i}^{\left(k\right)}<u)\;\;s.t.u_{i}^{\left(k\right)}\in T^{\left(k\right)}\}|,T^{\left(k\right)}=DomS^{\left(k\right)}\cup T^{\left(k-1\right)}$
*,*
$n_{T}^{\left(k\right)}$
*is the size of set*
$T^{\left(k\right)}$
*, and*
$t_{i}^{\left(k\right)}$
*is the number of individuals who dominate the set.*


A strength value is assigned such that each individual can be computed and multi-locally searched for the evolution of the Pareto optimal set intelligent optimization algorithm, and this value is the fitness determined by its advantages and disadvantages, namely,
(3)$$f_{i}^{\left(k\right)}=1+\sum_{u_{j}\in T^{\left(k\right)}\wedge u_{j}\wedge u_{i}}S_{i}^{\left(k\right)},$$
Here, the fitness of an individual is equal to the sum of all the external individuals that dominate it. In addition, the size of the set is chosen so that the individual is infinitely close to the fitness value.

### 3.2. Analytic Hierarchy Process

In the 1970s, Professor T. L. Saaty, an American operational research scientist, proposed a multi-objective decision analysis method called the AHP, which is mainly applied to decision-making problems under multiple evaluation criteria [24].

The modelling steps of the AHP are as follows:

(1) Establish a hierarchical structure model.

The hierarchical structure model is decomposed into the various factors that are contained in the problem, which form several levels from top to bottom according to different attributes [25]. The structure model framework includes the following levels: the highest level (the overall goal of the complex system); the middle level (the planning and decision-making, the measures and adopted policies, and the criteria for achieving the goals), and the lowest level (various strategies and constraints).

**Definition** **2.**
*Assuming that the set of evaluation indicators is *
$T=\{t_{1},t_{2},\ldots ,t_{n}\}$
*, the evaluation indicators of each factor are compared with a pair of importance degrees, and the determined value*
$\alpha_{ij}$
*is set to represent the importance degree of *
$t_{i}$
*. Then, the hierarchical structure model is as follows:*
(4)αij>0,αij=1αji,
*where*
$\alpha_{ij}$
*is the scale.*


Then, the judgment matrix *D* is as follows:(5)$$D=\left[\begin{array}{cccc} X_{11} & X_{12} & \cdots & X_{1n}\\ X_{21} & X_{22} & \cdots & X_{2n}\\ \vdots & \vdots & \vdots & \vdots \\ X_{m1} & X_{m2} & \cdots & X_{mn} \end{array}\right]=\left[\begin{array}{cccc} \frac{X_{1}}{X_{1}} & \frac{X_{1}}{X_{2}} & \cdots & \frac{X_{1}}{X_{n}}\\ \frac{X_{2}}{X_{1}} & \frac{X_{2}}{X_{2}} & \cdots & \frac{X_{2}}{X_{n}}\\ \vdots & \vdots & \vdots & \vdots \\ \frac{X_{n}}{X_{1}} & \frac{X_{n}}{X_{2}} & \cdots & \frac{X_{n}}{X_{n}} \end{array}\right],$$
It can be seen from the above that for the positive definite reciprocal judgment matrix *D*, the maximum characteristic root $\lambda_{\max}$ exists and is unique, and the weight $w_{i}$ is composed of positive elements of vectors and is unique.

(2) Construct a judgment matrix.

The multi-level system is divided into several hierarchical levels according to different goals and functions [26]. Among them, the judgment matrix is constructed with the pairwise comparison method and comparison scale at the lowest level.

The elements in each row of the judgment matrix are multiplied as follows:(6)$$M_{i}=\overset{n}{\underset{i=1}{\Pi }}\alpha_{ij},$$

$M_{i}$ is calculated by taking the n-th power root:(7)$$W_{i}=\sqrt[{n}]{M_{i}},$$

$W_{i}$ in the vector is normalized:(8)$$W_{i}=W_{i}/\sum_{i=1}^{n}W_{i},$$

Here, the values of vector $W_{i}$ are normalized for the consistency check.

(3) Determine the hierarchical order and perform a consistency check.

Here, the method determines the correlation degree between adjacent level elements in the above judgment matrix. Through the construction of two comparison judgment matrices and the mathematical method of matrix operations, the importance order of the related elements is determined for a certain element in the previous level.

**Definition** **3.**
*In checking the consistency of the judgment matrix, knowledge and experience can meet different conditions. The specific formula is as follows:*
(9)$$C_{R}=\frac{C_{i}}{R_{i}}=\frac{\lambda_{\max}-n}{(n-1)R_{i}},$$
*where*
$C_{i}$
*is the indicator of the consistency check,*
n
*is the order of the*
*judgment matrix, and*
$R_{i}$
*is the average value of the consistency check.*


(4) Determine the hierarchical total ranking and perform a consistency check.

The combination weight vector of each layer element is calculated and ranked by the formula of the combination consistency check. Therefore, it determines the importance degree of each element at the bottom of the hierarchical structure model. In the traditional AHP method, it is difficult to verify the consistency of the judgment matrix, and this matrix is greatly affected by expert knowledge and preferences, which limits its promotion and application [27]. The concept of Multi-indicator Entropy is proposed, and the indicator weight solution method solves the problems of the traditional AHP method. The evaluation entropy refers to the confusion degree of the evaluation value of each candidate C3DPS set in a comprehensive evaluation.

**Definition** **4.**
*Assuming that for a certain service request, the set of candidate C3DPSs is*
$CS=(S_{1},S_{2},\ldots ,S_{n})$
*and*
$QT_{11},QT_{12},\ldots ,QT_{1n}$
*are defined as the evaluation values of the candidate C3DPSs on the indicators, the evaluation entropy of the indicators*
$QT_{1}$
*is*
(10)$$Diff(QT_{1})=\frac{\sqrt{\frac{\sum_{i=1}^{n}{\left(QT_{1i}-\bar{QT_{1}}\right)}^{2}}{n-1}}}{QT_{1}},$$


The solution formula for index evaluation entropy is introduced into other secondary indicators of the QoS index, and $Diff(QT_{2})$,..., $Diff(QT_{i})$ are obtained; then, the relative weight corresponding to $QT_{x}\;\;(x=1,2,\ldots ,n)$ is
(11)$$W(QT_{x})=\frac{Diff(QT_{x})}{\sum_{i=1}^{n}Diff(QT_{x})},$$
where $W(QT_{x})$ is the weight value, $Diff(QT_{x})$ is the indicator evaluation entropy of $Diff(QT_{x})$, and $\sum_{i=1}^{n}Diff(QT_{x})$ is the sum of the entropy values of $QT_{x}\;\;(x=1,2,\ldots ,n)$.

Based on Pareto optimal theory, the improved AHP-TOPSIS evaluation model is used to objectively determine the weight of each evaluation set and index [28]. The closeness of the evaluation model is determined by analyzing and making decisions to obtain the optimal solution that brings about an overall similarity between the best alternative and the ideal scheme combined with the TOPSIS optimization method.

## 4. Mathematical Model of C3DPS Order Task Execution Evaluation Based on the AHP-TOPSIS Evaluation Model

In the process of multi-attribute decision-making, the weight of each attribute reflects the relative importance of the attributes, which directly affects the result of decision-making. Therefore, the weight of each attribute is one of the key issues in multi-attribute decision-making that determines the results of decision-making. Aiming to solve a multi-attribute decision-making problem in which the attribute weight information is determined and the attribute value is an intuitionistic fuzzy number, a decision analysis method for the AHP-TOPSIS evaluation model is proposed. This analysis method is widely used in multi-attribute decision-making problems. The AHP-TOPSIS comprehensive evaluation model is a comprehensive evaluation and optimization method that combines the AHP and TOPSIS. The weight vector of each evaluation indicator is objectively determined and calculated to achieve comprehensive superiority by this evaluation model [29]. The basic principle of TOPSIS is to sort the evaluation objects by the distance between the fuzzy positive ideal solution and its similarity to fuzzy comprehensive attributes in the multi-objective decision-making problem.

### 4.1. Establish an Initial Evaluation Matrix A

**Definition** **5.**
*Suppose that a set*
$A=\{A_{1},A_{2},\ldots ,A_{m}\}$
*of schemes is composed of m schemes*
$A_{1},A_{2},\ldots ,A_{m}$
*and that each scheme also corresponds to a number of evaluation indicators*
$X_{1},X_{2},\ldots ,X_{n}$
*. The set of evaluation indicators is*
$X=\{X_{1},X_{2},\ldots ,X_{n}\}$
*. Then, the initial evaluation indicator matrix can be expressed as follows:*
(12)$$A={\left(X_{ij}\right)}_{m\times n}=\left[\begin{array}{cccc} X_{11} & X_{12} & \cdots & X_{1n}\\ X_{21} & X_{22} & \cdots & X_{2n}\\ \vdots & \vdots & \vdots & \vdots \\ X_{m1} & X_{m2} & \cdots & X_{mn} \end{array}\right],$$
*where*
$X_{ij}$
*is the*
j-th
*evaluation indicator in the*
j-th
*scheme.*


### 4.2. Establish a Weighted Standardized Decision Matrix

The evaluation indicators can be divided into two categories: consumption indicators and profit indicators. The higher the profit indicator is, the lower the consumption indicator is. These are two kinds of indicators that can also be divided into measurement indicators and nonmeasurement indicators with different dimensions and dimensional units [30]. On this basis, it is necessary to strictly define the meaning of indicators and provide reference standards, which will eliminate the incontestability of the resulting indicators and carry out dimension normalization of the evaluation indicators in the evaluation of nonmetric indicators.

The standardized decision calculation method for the initial evaluation matrix is described below.

(1) The consumption indicator is
(13)$$b_{ij}=\frac{X_{ij}-\min_{j}(X_{ij})}{\max_{j}(X_{ij})-\min_{j}(X_{ij})},$$
where $X_{ij}-\min_{j}(X_{ij})$ is the difference between the evaluation indicator value and the minimum evaluation indicator value in the initial evaluation matrix; $\max_{j}(X_{ij})-\min_{j}(X_{ij})$ is the difference between the maximum value and the minimum value in the initial evaluation matrix.

(2) The profit indicator is
(14)$$b_{ij}=\frac{\max_{j}(X_{ij})-X_{ij}}{\max_{j}(X_{ij})-\min_{j}(X_{ij})},$$
where $\max_{j}(X_{ij})-X_{ij}$ is the difference between the maximum evaluation indicator value and the evaluation indicator value in the initial evaluation matrix, and $\max_{j}(X_{ij})-\min_{j}(X_{ij})$ is the difference between the maximum value and the minimum value in the initial evaluation matrix.

According to the life cycle evaluation indicator system of C3DPSs, a standardized decision matrix for multi-attribute decision-making is constructed. According to the accumulation process of the performance indicator MOpt(T, Q, Mat, R, Rl, Flex, C, Ft, SF, Sa), the objective function of multi-objective optimization is determined, and the performance of the C3DPSs can be evaluated through four sub-evaluation indicator systems [31]. Assuming that the service resources of C3DPSs are evaluated, $ORS=\{r_{1},r_{2},\ldots ,r_{n}\}$ will be selected as a candidate set of service resources in the C3DP order execution process. A standardized decision matrix for multi-attribute decision-making is shown in Table 1.

Here, $\left(\mu_{i}^{T},\nu_{i}^{T}\right)$, $\left(\mu_{i}^{Q},\nu_{i}^{Q}\right)$, $\left(\mu_{i}^{Mat},\nu_{i}^{Mat}\right)$, $\left(\mu_{i}^{R},\nu_{i}^{R}\right)$, $\left(\mu_{i}^{Rl},\nu_{i}^{Rl}\right)$, $\left(\mu_{i}^{Flex},\nu_{i}^{Flex}\right)$, $\left(\mu_{i}^{C},\nu_{i}^{C}\right)$, $\left(\mu_{i}^{Ft},\nu_{i}^{Ft}\right)$, $\left(\mu_{i}^{SF},\nu_{i}^{SF}\right)$, and $\left(\mu_{i}^{Sa},\nu_{i}^{Sa}\right)$ are, respectively, intuitionistic fuzzy numbers representing the duration time T, cost C, matching degree Mat, service response R, quality Q, reliability Rl, service fault tolerance Ft, flexibility Fl, safety SF, and customer satisfaction Sa. For the performance evaluation of the C3DPSs life cycle evaluation indicator system, multi-attribute decision-making is standardized as a decision-making matrix.

(3) Establish a weighted standardized decision matrix R.

The weighted standardized decision matrix R is multiplied by the column vector of the matrix X with the weight values determined by the AHP method, and the weighted standardized decision matrix R can be obtained as
(15)$$R={\left(r_{ij}\right)}_{m\times n}=\left[\begin{array}{cccc} w_{1}b_{11} & w_{2}b_{12} & \cdots & w_{n}b_{1n}\\ w_{1}b_{21} & w_{2}b_{22} & \cdots & w_{n}b_{2n}\\ \vdots & \vdots & \vdots & \vdots \\ w_{1}b_{m1} & w_{2}b_{m2} & \cdots & w_{n}b_{mn} \end{array}\right],$$
Here, w is a coefficient of the matrix.

(4) Calculate the closeness $B_{i}^{+}$ of the evaluation objects.

The ideal solution of the profit indicator set $J_{1}$ is the maximum value of the row vector, and the negative ideal solution is the minimum value of the row vector [32]. The ideal solution of the consumption indicator set $J_{2}$ is the opposite.
(16)$$\{\begin{array}{c} R^{+}=\{(\max(w_{n}b_{mn}|m\in J_{1})),(\min(w_{n}b_{mn}|m\in J_{2}))\}\\ R^{-}=\{(\min(w_{n}b_{mn}|m\in J_{1})),(\max(w_{n}b_{mn}|m\in J_{2}))\} \end{array}.$$

The matrix of the evaluation object and ideal solution is as follows:(17)$$\{\begin{array}{c} D^{+}=\sqrt{\sum_{j=1}^{n}{\left(r_{ij}-r_{j}^{+}\right)}^{2}}\\ D^{-}=\sqrt{\sum_{j=1}^{n}{\left(r_{ij}-r_{j}^{-}\right)}^{2}} \end{array},$$
where $D^{+}$ and $D^{-}$ are the distances between the evaluation object and the positive and negative ideal solutions, respectively; $r_{j}^{+}$ and $r_{j}^{-}$ are the elements corresponding to $R^{+}$ and $R^{-}$.

The formula for the closeness of the evaluation object is as follows:(18)$$B_{i}^{+}=\frac{D^{-}{}_{i}}{D^{+}{}_{i}+D^{-}{}_{i}},\;\;0\leq D^{+}{}_{i}\leq 1,$$

(5) Construct an AHP-TOPSIS comprehensive evaluation model.

The evaluation matrix is constructed from the proximity analysis of the TOPSIS method, and the result vector F of the AHP-TOPSIS comprehensive evaluation is as follows:(19)F=W×B,
In the formula, B is an evaluation matrix formed from the closeness value of each evaluation object, and W is the weight calculated by the analytic hierarchy process.

## 5. The Solution of the C3DPS Quality Evaluation Model

### 5.1. Hybrid Multi-Objective Particle Swarm Optimization (PSO) Algorithm Based on the Baldwin Effect (BM-MOPSO)

BM-MOPSO is an intelligent algorithm based on multi-objective PSO that combines Baldwin’s learning strategy idea, a population global target value scalar parameter, a scalar parameter for the population global objective value, intuitionistic fuzzy membership, and a ranking method. In the application of the hybrid multi-objective PSO algorithm, it solves the above-mentioned C3DPS quality evaluation problem [33]. This is the key to the problem of adopting the learning of the Baldwin effect within a certain period so that the global search and learning strategies can ensure the interactive operation of the algorithm in a fixed period.

(1) Baldwin effect learning strategy.

As a learning method, the Baldwin effect learning strategy can effectively reduce the selection pressure. This not only affects the characteristics of the search space but also increases the polymorphism of the genetic process and transforms the shape of the dominant search space. A local search can also be carried out based on the Baldwin effect that improves the nondominated solution of PSO [34]. The mathematical formula of this learning function is as follows:(20)$$Y_{ij}=\{\begin{array}{c} p_{zj}+k\times (p_{ij}-p_{zj}),\;p_{zj}<p_{ij}\\ p_{zj}+k\times (p_{zj}-p_{ij}),\;\;p_{ij}<p_{zj}\\ p_{ij},\;\;\;\;Otherwise \end{array},$$
where i,z={1,2,…,m}, m is the number of particles in the swarm, j={1,2,…,n}, and n is the dimension of the particle swarm.

If $p_{ij}$ is greater than $p_{zj}$, particles $p_{zj}$ learn from particles $p_{ij}$; if $p_{zj}$ is greater than $p_{ij}$, particles $x_{ij}$ learn from particles $x_{zj}$; if neither is dominated by the other, the particles do not learn.

(2) Local search strategy based on the BM-MOPSO algorithm.

When the particle swarm performs a local search for the Baldwin effect, one of the following situations will occur:

(1) When searching the initial points, most particles are far away from the Pareto frontier in the space, and it is easy to find the dominant solution in the region, which leads to high search efficiency. At a certain time, the optimal solution set of the population is sorted by Pareto dominance, and the particles are dynamically updated to the Pareto solution set. At the same time, the method learns from the dominant solution so that the particle swarm can more quickly approach the Pareto frontier.

(2) If the local search times of PSO converge to the threshold flag, the obtained solutions are all nondominated solutions, which indicates that the population has fallen into a local extreme point at this time. The direction of the nondominated solutions is
(21)$$d=\sum_{i=1}^{\mathrm{flag}}\frac{x_{i}-x_{0}}{\left||x_{i}-x_{0}|\right|},$$
where $x_{i}(i=1,2,\ldots ,\mathrm{flag})$ is the i−th nondominated solution in the local search and $x_{i}-x_{0}$ is the distance from the local extreme point to the initial value in the local search.

(3) If the above two conditions are satisfied, then the number d of $x_{i}$ and the nondominated solution q (q = 3) satisfy one of the following conditions:

If d<q, the method learns from the dominating solution so that the particle swarm can more quickly approach the Pareto frontier.

If d≥q, the search direction changes, and the PSO can be guided and diffused in the optimization direction.

The local search method of the BM-MOPSO algorithm is as follows:

**Step 1**: The parameters are initialized; that is, the initial position of PSO is $\left(x_{0},y_{0}\right)$, its iteration number is i=0, and the direction of the nondominated solution is d=0.

**Step 2**: In the initial position of PSO $\left(x_{0},y_{0}\right)$, a dominating solution is randomly selected and marked as $y_{1}$.

**Step 3**: If $y_{1}<y_{0}$, the initial position of PSO is locally searched by the Baldwin effect learning strategy and is set to $x_{\mathrm{new}}=y_{0}+k\times (y_{1}-y_{0})$ (k is the number of executions of the Baldwin effect learning strategy); then, the method goes to **step 5**. If $y_{0}<y_{1}$, then the Pareto solution set is dynamically updated. At the same time, the method learns from the dominant solution. The particle swarm can move closer to the Pareto frontier, ensuring that the individual particle swarm will have this learning ability. If $x_{\mathrm{new}}=y_{1}+k\times (y_{0}-y_{1})$, the method goes to **step 5**; otherwise, it goes to **step 4**.

**Step 4**: If d++, then PSO calculates the direction d of the nondominated solution. If d≥q, it reinforces learning to calculate the optimal step size s and explore the optimal position $x_{\mathrm{new}}=y_{0}+d\times s$ and then goes to **step 5**; otherwise, it goes to **step 2**.

**Step 5**: If i≤flag, the method goes to **step 2**; otherwise, this particle swarm population falls into a local extreme point, and is the next initial value of particle swarm $x_{0}$.

### 5.2. The Basic Process of the Multi-Objective Particle Swarm Optimization Algorithm Based on the Baldwin Effect

The basic process of multi-objective BM-MOPSO is shown in Figure 2. The algorithm steps are analyzed below.

(1) Particle initialization.

First, the maximum number of iterations, number of independent variables of the objective function, maximum velocity of particles, and position information are set randomly in the velocity interval and search space to obtain a one-to-one mapping between the service resources and particles, that is, to perform particle initialization.

Assume that each particle is an optional combination of service resources, where a service resource is selected from each candidate service set list $CRS_{i}$ to form a combination of service resources [35]. The initial scale of the particles is the n-population of the feasible dimension space $S=\{p_{1},p_{2},\ldots ,p_{n}\}$, and each scale of particles corresponds to an optional combination of the number of service resources.

In the same way, the order of particles mapped one by one is. A mapping example of the composition of service resources and particles is shown in Figure 3.

For example, let the combination of C3DP optional service resources be $\left\{CRS_{1}^{5},CRS_{2}^{4},CRS_{3}^{1},CRS_{4}^{5}\right\}$, which means that a C3DPS resource is selected by the serial atomic task sequence $OR_{1}$ to be the fifth service resource $CRS_{1}^{5}$ in the set of service resources $CRS_{1}$. The serial atomic task sequence $OR_{2}$ is the fourth service resource $CRS_{2}^{4}$ in the set of service resources $CRS_{2}$ [36].

(2) Set of fitness functions.

From the above objective function MOpt(OR), the optimal value of each index set is selected by the evaluation of the C3DP order task in the feasible solution domain of the service resource candidate service subset. Therefore, the moderation function of the BM-MOPSO algorithm is calculated as follows:(22)$$\begin{array}{l} fitness(p^{i})=w_{1}{\left(\sum_{j=1}^{n}T(CRS_{j}^{p_{j}^{i}})\right)}^{-1}+w_{2}{\left(\sum_{j=1}^{n}Q(CRS_{j}^{p_{j}^{i}})\right)}^{-1}+w_{3}\prod_{i=1}^{n}{\left(\sum_{j=1}^{n}Mat(CRS_{j}^{p_{j}^{i}})\right)}^{-1}+w_{4}\prod_{i=1}^{n}{\left(\sum_{j=1}^{n}R(CRS_{j}^{p_{j}^{i}})\right)}^{-1}\\ \begin{array}{cccc} & & & w_{5}\prod_{i=1}^{n}{\left(\sum_{j=1}^{n}Flex(CRS_{j}^{p_{j}^{i}})\right)}^{-1}+w_{6}{\left(\sum_{j=1}^{n}C(CRS_{j}^{p_{j}^{i}})\right)}^{-1}+w_{7}\prod_{i=1}^{n}{\left(\sum_{j=1}^{n}Ft(CRS_{j}^{p_{j}^{i}})\right)}^{-1}+w_{8}\prod_{i=1}^{n}{\left(\sum_{j=1}^{n}Sa(CRS_{j}^{p_{j}^{i}})\right)}^{-1} \end{array} \end{array}$$

In the formula, $w_{1}$, $w_{2}$, $w_{3}$, $w_{4}$, $w_{5}$, $w_{6}$, $w_{7}$ and $w_{8}$ are the weight ratios of C3DP equipment service resources. The larger the fitness function $fitness(p^{i})$ is, the better the particle $p^{i}$ is.

(3) The range of the particle dimension and moderate function.

In the BM-MOPSO algorithm, the learning strategy based on the Baldwin effect is an iterative process of learning and evolution that balances the relationship between global search and local search. To improve the nondominated solution of particle swarms, an individual particle learns in the same environment to achieve stronger survival adaptability [37].

After completing the Baldwin effect learning operation, the value of a certain dimension of the particle is larger than the value range, so it is necessary to calculate the particle swarm with the extreme value standardization method, which is used if the value is larger than the value range.

Therefore, the value range of each dimension of the particles is a discrete value range $\left\{p_{k}^{j}:1\leq p_{k}^{j}\leq K_{j}\right\}$. After the Baldwin effect learning operation is completed, the dimension value of the particles is greater than the value range, and the extreme value normalization method is used for the particle swarm in this excessive value range. If $p_{k}^{j}>K_{j}$, the value is set to $p_{k}^{j}=K_{j}$.

The optimal position of each individual $p_{i}$ of the particle swarm is now $x_{i}=(x_{i1},x_{i2},\ldots ,x_{ij})$. The functional generalized derivative f(x) representing the feedback information in the learning and evolution process of the BM-MOPSO algorithm is calculated, which is defined as follows:(23)$$\frac{Df(p_{i})}{Dx_{ik}}=\frac{f(x_{i1},\ldots ,x_{ik}+△x_{ik},\ldots ,x_{ij})-f(x_{i1},\ldots ,x_{ik},\ldots ,x_{ij})}{△x_{ik}},$$
Here, the individual particle $p_{i}$ yields $p_{i}{}^{\prime }$ when the learning strategy based on the Baldwin effect is carried out. For each dimension $x_{ik}$(k=1,2,…,j) of the position vector of individual, the formula is as follows:(24)$$\begin{array}{l} \underset{|p_{i}-p_{i}^{\prime }|\leq \varepsilon }{\min}f(p_{i}^{\prime })=\underset{|p_{i}-p_{i}^{\prime }|\leq \varepsilon }{\min}f(x_{i1}^{\prime },x_{i2}^{\prime },\ldots ,x_{ij}^{\prime })\\ \;\;\;\;\;\;\;\;\begin{array}{ccc} & = & \underset{N_{i1},N_{i2},N_{ij}\in Z^{+}}{\min} \end{array}f(x_{i1}+\frac{\lambda_{i1}N_{i1}Df(p_{i})}{Dx_{i1}},\ldots ,x_{ij}+\frac{\lambda_{ij}N_{ij}Df(p_{i})}{Dx_{ij}}) \end{array}$$
where $\lambda_{ik}$ and $N_{ik}$ are the parameters for adjusting the step length. The integer $N_{ik}$ is the number of individuals in the population particle swarm $p_{i}$, and their initial values are all the same; $N_{ik}+1\to N_{ik}$ and $\lambda_{ik}$ is the value of the updated particle position within the feasible region $\left[x_{L},x_{R}\right]$ [38]. Then, $\lambda_{i}$ is the quotient of the range distance and the maximum iteration number:(25)$$\lambda_{i}=\frac{\left|x_{L}-x_{R}\right|}{T_{\max}},$$

The steps of the algorithm are as follows:

**Step 1**: Initialization. This includes all parameters of the particles, such as the initial position and velocity. The individual optimal position of a particle is defined as the current position, and the global optimal position is the optimal position of all particles. The initial position of each particle i is $x_{i}=(x_{1},x_{2})$, the speed is $\overset{\to G}{v_{i}}=(v_{1},v_{2})$ (i=1,…,N), the number of particles is $n_{p}$, the number of iterations is $n_{\max}$, and the initial solutions $p^{0}$ are randomly generated by $n_{p}$.

**Step 2**: Calculate the fitness function value of each particle. That is, the fitness function value $fitness(p^{i})$ is calculated by the functional generalized derivative f(x) when the coordinate $x_{i}$ takes the coordinates into the generalized derivative.

**Step 3**: Determine whether to perform local search. If the current iteration number meets the conditions for iteration termination (<13 of the total number of iterations), then local search is not performed and the algorithm moves to **step 4**; otherwise, **step 5** is performed.

**Step 4**: Perform a global search. If the number of consecutive iterations with no updates meets the preset threshold, each particle will calculate the update speed and position in the global optimal solution according to the basic formula; otherwise, the algorithm moves to **step 6**;

**Step 5**: Perform a local search. The particles perform the local search based on the Baldwin effect with probability $P_{k}$, and the algorithm moves to **step 6**;

**Step 6**: Non-uniform mutation. Assuming that there is a particle $p^{t}=(v_{1},\ldots ,v_{j})$ in the t−th generation, a random variable $v_{k}$ is selected and outputs a number between 0 and 1, and a non-uniform mutation operator is locally mutated by the particle for the next generation $p^{t+1}=(v_{1},\ldots ,v_{k}^{\prime },\ldots ,v_{j})$. The particle swarm will become more stable.

**Step 7**: Repeat **step 2** to **step 6** until the current iteration number meets the conditions for termination; otherwise, continue to update $x_{i}$ and $v_{i}$.

**Step 8**: After execution, output the calculation result.

## 6. Example Simulation

To ensure the preciseness of the data, the case data were sampled from a CMfg platform developed by a 3D printing company in Wuhan city, which is a smart C3DPS platform that integrates modelling design and 3D printing. It integrates various kinds of C3DPSs of multiple fields and types and provides network access to different types of 3D printing equipment. It also performs online real-time data collection. Here, the real data of different 3D printing devices in the platform were selected for the example simulation.

### 6.1. C3DPSs Modeling

Taking the construction of C3DPS network as an example, a computer program is clustered to the structure of the C3DPS network. The relevant basic data of the network must be described as a graph in the form of a database table, and the database table is the structure of 3D printing service network. The simulation process of the C3DPS network is divided into the following steps:

Firstly, it expresses the information of C3DPSs. As the assembly resources of personalized design products have many types and different capabilities, the C3DPS template is expressed in XML format.

For example, some of the attributes of Line 3 of No.6 Studio of Urui 3D Printing include 3D printing service mode, 3D printing specifications and dimensions, printing accuracy, maximum printing speed, 3D printing equipment model, and printing type. The node relationship is shown in Figure 4. As shown in Figure 5, it is a C3DPS expressed in XML format. At the same time, the XML format is expressed to the logical relationships among resources, attributes, and attribute values.

For example, some attributes of Line 3 of No.6 Studio of Yourui 3D Printing include 3D printing service mode, 3D printing specifications and dimensions, printing accuracy, maximum printing speed, 3D printing equipment model, and printing type, and their node relationships are shown in Figure 4. As shown in Figure 5 and Figure 6, it is a C3DPS in XML format. At the same time, the C3DPS also uses XML format to express the logical relationship among resources, attributes, and attribute values.

Secondly, it determines the service node. Here, as the research object, 10 C3DPS providers is selected and established to the basic information table of the node, as shown in Table 2.

Here, a program is incorporated in the software to cluster the structure of the C3DPS network, so as to describe the triple data of the cultural creative product personalized design product as a graph in the form of a database table and generate a complex network topology of the cultural creative product personalized design. Among them, the basic information of portrait 3D printing node and data attributes of C3DPS node are shown in Table 3.

Thirdly, it determines the alternative relationship between the candidate C3DPS providers (that is, there is a competitive relationship by the same or similar services providers) and creates a node connection relationship table.

Because of the different comprehensive service capabilities of each C3DPS provider, it is necessary to investigate each service node. For example, Jiayi Hi-Tech 3D scanner service provider provides 3D scanner outsourcing services, and the Worry 3D scanner service providers also provide 3D scanner outsourcing services. The output service capabilities of both are the same, so their similar weight is 1. After the data of all C3DPSs are summarized and sorted, the node connection relationship table is generated according to the form of a database table, as shown in Table 4.

Finally, the basic information table and connection relationship table of C3DPS node are imported into gephi-0.8.1 software for data analysis. The topology of C3DPS (S_Net) network is shown in Figure 7. It is composed of 33 nodes and 46 edges. Among them, the same color of nodes means that the types of C3DPSs are similar. The thickness of the edge represents the similarity between the two C3DPSs, that is, the service capability is replaceable.

### 6.2. Simulation Environment

To verify the reliability and universality of the cloud service evaluation based on the hybrid multi-objective BM-MOPSO evaluation model, a verification application in C3DPS creative and innovation product development was used. Its simulation environment was as follows:Windows 7 operating system;Intel (R) Core (tm) i5-4210H 2.90 GHZ CPU;8G memory.

The experimental environment was as follows:

At run time, the population size was 10, the maximum number of iterations was 100, the target weights were $Q_{1}$ and $Q_{2}$, and the values were 0.7 and 0.3, respectively. The range of moderate function deterioration was set to −0.1 for the moderate functions $w_{1}=0.20$, $w_{2}=0.20$, $w_{3}=0.10$, $w_{4}=0.05$, $w_{5}=0.10$, $w_{6}=0.20$, $w_{7}=0.05$, and $w_{8}=0.10$.

According to the above multi-objective BM-MOPSO, the evaluation data and weight value of each candidate 3D printing device were inputted and simulated in MATLAB. Each algorithm was tested independently for each test function fewer than 30 times. The convergence characteristics of the algorithms are shown in Figure 8 and Figure 9:

Here, it is a flow chart of cloud service evaluation implementation based on the hybrid multi-objective BM-MOPSO evaluation model.

### 6.3. Analysis of Hybrid Multi-Objective BM-MOPSO

The Knowledge module is responsible for the management and scheduling of various types of knowledge on service resources; the Coordination module creates links between various coordination methods and performs operation monitoring and coordination management in the cloud service evaluation model based on the hybrid multi-objective BM-MOPSO evaluation model. Figure 10 shows the evaluation process for cloud services based on the hybrid multi-objective BM-MOPSO evaluation model.

A user proposes a complex manufacturing task that decomposes into six sub-task nodes $T=(T_{1},T_{2},T_{3},T_{4},T_{5},T_{6})$. After entering the task information into the platform, it is preliminarily matched to several 3D printing equipment resources that meet the user’s needs. A candidate set of 3D printing equipment resources is established, that is, $S=(S_{1},S_{2},S_{3},S_{4},S_{5},S_{6})$. Among them, task node $S_{1}$ is matched with three candidate resources, which can be expressed by $S_{1}=(S_{11},S_{12},S_{13})$. Each candidate resource is shown in Table 5.

Where $S=(S_{1},S_{2},S_{3},S_{4},S_{5},S_{6})$ means Services = (Service 1 (Point cloud data processing), Service 2 (3D Reverse Engineering), Service 3 (3D CAD modeling and design), Service 4 (Slicing procedures for layered manufacturing technique), Service 5 (simulation analysis), and Service 6 (3D printing and post-processing)}. $S_{1}=(S_{11},S_{12},S_{13})$ means three Candidates for 3D printing device resources.

According to the different 3D printing equipment resources, the evaluation indicators are optimized and quantified from the original data of each 3D printing equipment resource. The form of the data set is as follows:(26)$$S_{1}=\left[\begin{array}{ccccc} q_{c}(S_{i1}) & q_{t}(S_{i1}) & q_{r}(S_{i1}) & q_{u}(S_{i1}) & q_{e}(S_{i1})\\ q_{c}(S_{i2}) & q_{t}(S_{i2}) & q_{r}(S_{i2}) & q_{u}(S_{i2}) & q_{e}(S_{i2})\\ q_{c}(S_{i3}) & q_{t}(S_{i3}) & q_{r}(S_{i3}) & q_{u}(S_{i3}) & q_{e}(S_{i3})\\ \vdots & \vdots & \vdots & \vdots & \vdots \\ q_{c}(S_{ij}) & q_{t}(S_{ij}) & q_{r}(S_{ij}) & q_{u}(S_{ij}) & q_{e}(S_{ij}) \end{array}\right]$$
Here, the unit of $q_{c}(S_{ij})$ is “Yuan”, and the unit of $q_{t}(S_{ij})$ is “hour”.

The above evaluation data of each candidate 3D printing equipment resource are specifically expressed as follows:(27)$$\begin{array}{cc} S_{1}=\left[\begin{array}{ccccc} 1240 & 35 & 0.76 & 0.79 & 78\\ 1300 & 55 & 0.18 & 0.48 & 94\\ 1750 & 54 & 0.44 & 0.64 & 98 \end{array}\right] & S_{2}=\left[\begin{array}{ccccc} 980 & 42 & 0.79 & 0.75 & 20\\ 2200 & 59 & 0.27 & 0.65 & 34\\ 1700 & 61 & 0.65 & 0.16 & 70\\ 3200 & 59 & 0.11 & 0.49 & 30 \end{array}\right]\\ S_{3}=\left[\begin{array}{ccccc} 1400 & 49 & 0.95 & 0.34 & 53\\ 1300 & 58 & 0.58 & 0.22 & 64\\ 2700 & 67 & 0.75 & 0.25 & 87 \end{array}\right] & S_{4}=\left[\begin{array}{ccccc} 1850 & 30 & 0.50 & 0.69 & 49\\ 3300 & 34 & 0.89 & 0.95 & 40\\ 2000 & 24 & 0.54 & 0.13 & 68 \end{array}\right]\\ S_{5}=\left[\begin{array}{ccccc} 1600 & 55 & 0.14 & 0.25 & 49\\ 2600 & 36 & 0.84 & 0.25 & 68\\ 1850 & 37 & 0.81 & 0.24 & 75\\ 900 & 44 & 0.92 & 0.35 & 53 \end{array}\right] & S_{6}=\left[\begin{array}{ccccc} 750 & 43 & 0.19 & 0.25 & 35\\ 2700 & 37 & 0.61 & 0.47 & 67\\ 1800 & 38 & 0.35 & 0.83 & 59\\ 1000 & 68 & 0.58 & 0.54 & 76 \end{array}\right], \end{array}$$

Assuming that the particle population is 14 and the number of iterations is 50, the weight values of the evaluation parameters in the fitness function are, respectively, $\alpha_{1}=0.2$, $\alpha_{2}=0.3$, $\alpha_{3}=0.2$, $\alpha_{4}=0.1$, and $\alpha_{5}=0.2$.

According to the above multi-objective BM-MOPSO, the evaluation data and weight value of each candidate 3D printing device are inputted and simulated in MATLAB. The simulation results are shown in Figure 11, in which the vertical axis is the fitness function value and the horizontal axis is the population number.

It can be seen from Figure 6 that when the population number is 14, the fitness is lowest, the optimal combination scheme is $S=(S_{11},S_{21},S_{31},S_{42},S_{54},S_{63})$, and the fitness function value is 3.9370. That is, the optimal combination of 3D printing equipment resources is Yourui 3D printing hw-602, WINBO WB-SH105 and Qiaoyi workshop SLM 500, Flashcast Technology Explorer, Jiayi Hi-Tech JOYE-1212E, and Yunle Design Studio.

## 7. Conclusions

The global economy is transforming from a product economy to a service economy. Manufacturing and services are gradually merging. Individual enterprises pay close attention to their own core business. By providing manufacturing services, they can increase the value for stakeholders in the manufacturing value chain so that these individual enterprises will be more closely coordinated with each other. The C3DP model is a new service model that supports multi-user collaboration, and it is also an application of cloud manufacturing in the field of 3D printing. C3DPS modelling is the basis of C3DPS supply-and-demand matching; that is, this relationship between order tasks and services provides an effective way to match similar elements in the C3DPS platform. A large number of C3DPSs are aggregated according to certain rules and form a multi-complex 3D printing service network.

This paper formally describes C3DPSs, proposes a QoS acquisition and calculation method based on a mutual evaluation mechanism under the CMfg model, and establishes a C3DPS trust evaluation model based on service matching and global trust. The genetic algorithm optimizes the combination of C3DPSs so that it meets the requirements. Active intelligent rent-seeking for C3DPSs will be the next research direction.

## Figures and Tables

**Figure 1 micromachines-12-00801-f001:**
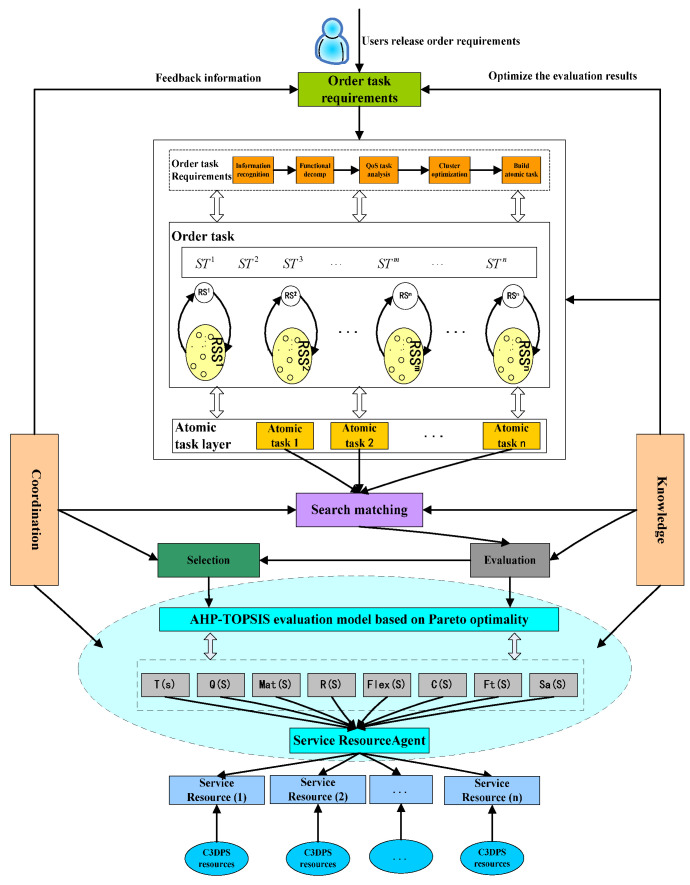
Framework of the evaluation method of the C3DPS order task execution process based on the AHP-TOPSIS optimal set algorithm and Baldwin effect.

**Figure 2 micromachines-12-00801-f002:**
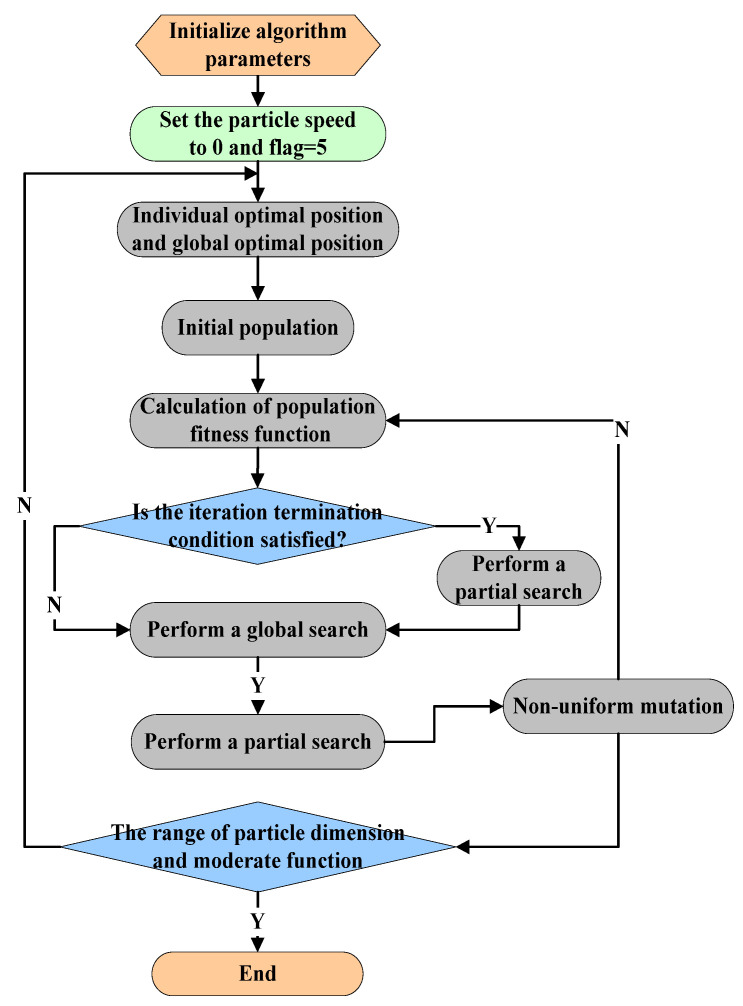
The basic process of the multi-objective BM-MOPSO algorithm.

**Figure 3 micromachines-12-00801-f003:**
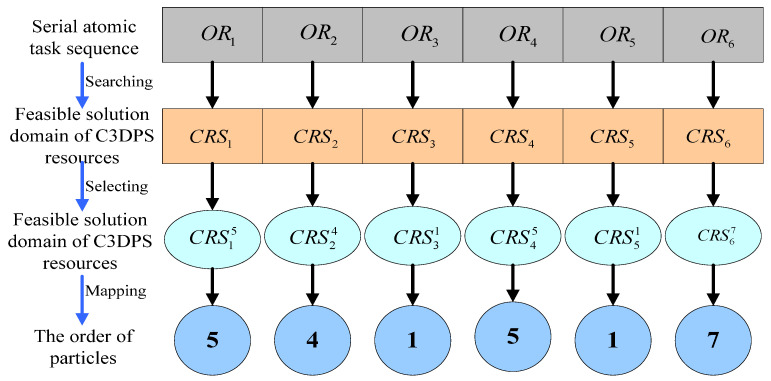
Mapping example of the composition of service resources and particles.

**Figure 4 micromachines-12-00801-f004:**
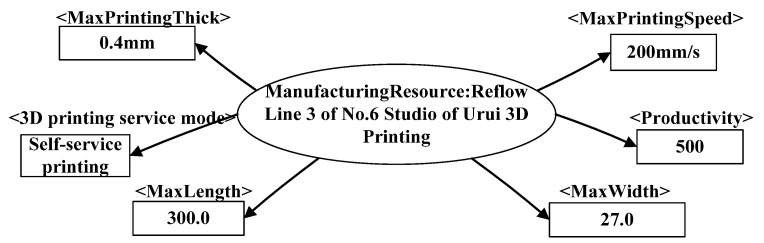
The relationship diagram of C3DPS node.

**Figure 5 micromachines-12-00801-f005:**
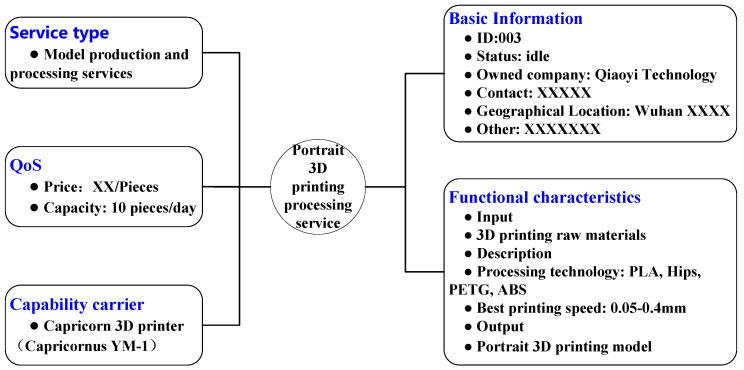
The description diagram of C3DPS.

**Figure 6 micromachines-12-00801-f006:**
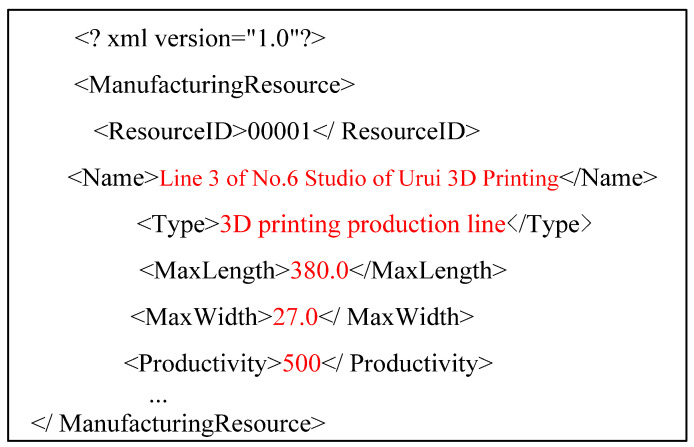
The Partial XML information graph of C3DPS template.

**Figure 7 micromachines-12-00801-f007:**
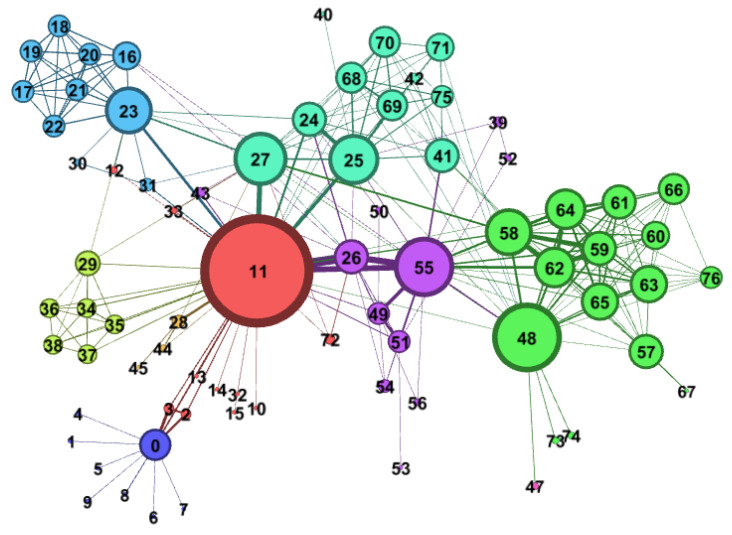
The network graph of C3DPS Network (S_Net).

**Figure 8 micromachines-12-00801-f008:**
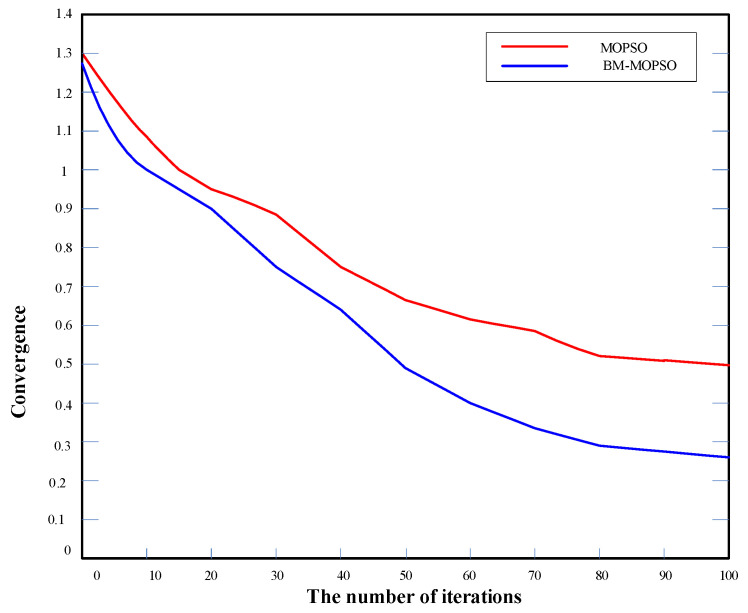
Convergence characteristics.

**Figure 9 micromachines-12-00801-f009:**
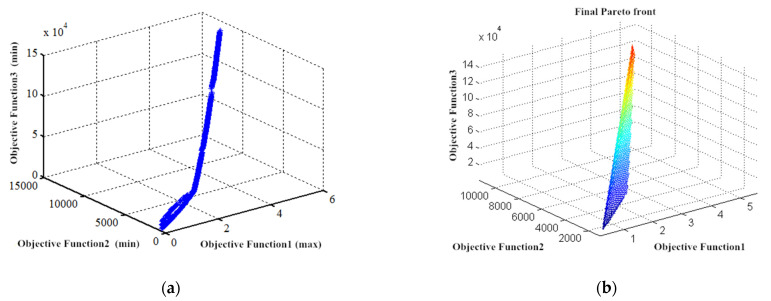
Simulation results of the algorithm. (**a**) Iterative simulation of the objective function; (**b**) The final Pareto frontier based on the improved BM-MOPSO algorithm. A caption on a single line should be centered.

**Figure 10 micromachines-12-00801-f010:**
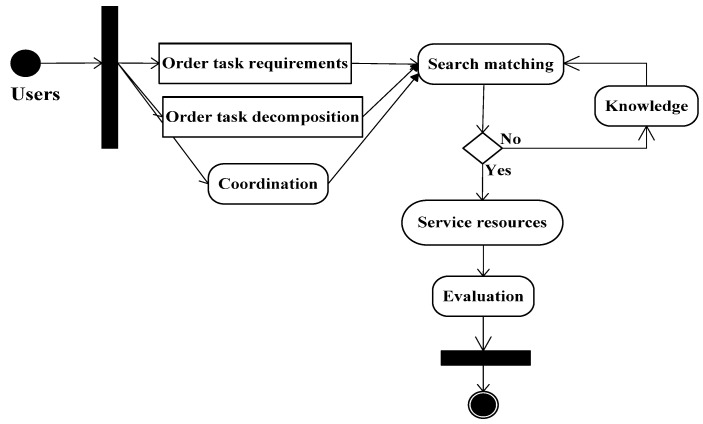
Flow chart of cloud service evaluation implementation based on the hybrid multi-objective BM-MOPSO evaluation model.

**Figure 11 micromachines-12-00801-f011:**
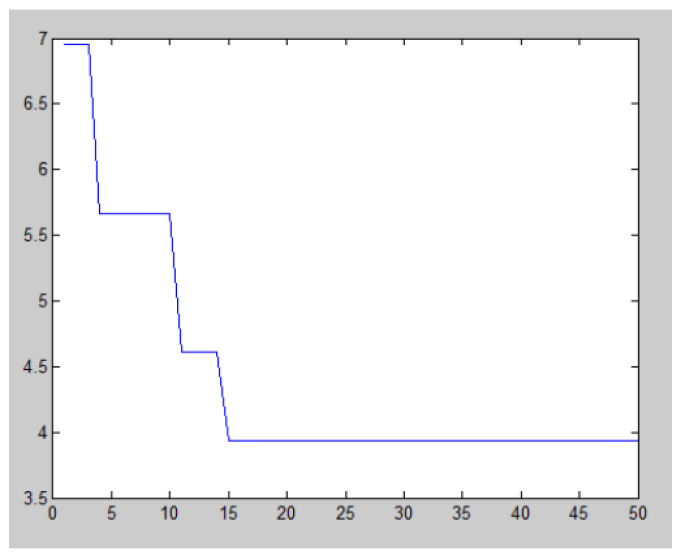
Convergence curve of the fitness function.

**Table 1 micromachines-12-00801-t001:** Standardized decision matrix for multi-attribute decision-making.

	$r_{1}$	$r_{2}$	…	$r_{i}$	…	$r_{n}$
T	$\left(\mu_{1}^{T},\nu_{1}^{T}\right)$	$\left(\mu_{2}^{T},\nu_{2}^{T}\right)$	…	$\left(\mu_{i}^{T},\nu_{i}^{T}\right)$	…	$\left(\mu_{n}^{T},\nu_{n}^{T}\right)$
Q	$\left(\mu_{1}^{Q},\nu_{1}^{Q}\right)$	$\left(\mu_{2}^{Q},\nu_{2}^{Q}\right)$	…	$\left(\mu_{i}^{Q},\nu_{i}^{Q}\right)$	…	$\left(\mu_{n}^{Q},\nu_{n}^{Q}\right)$
Mat	$\left(\mu_{1}^{Mat},\nu_{1}^{Mat}\right)$	$\left(\mu_{2}^{Mat},\nu_{2}^{Mat}\right)$	…	$\left(\mu_{i}^{Mat},\nu_{i}^{Mat}\right)$	…	$\left(\mu_{n}^{Mat},\nu_{n}^{Mat}\right)$
R	$\left(\mu_{1}^{R},\nu_{1}^{R}\right)$	$\left(\mu_{2}^{R},\nu_{2}^{R}\right)$	…	$\left(\mu_{i}^{R},\nu_{i}^{R}\right)$	…	$\left(\mu_{n}^{R},\nu_{n}^{R}\right)$
Rl	$\left(\mu_{1}^{Rl},\nu_{1}^{Rl}\right)$	$\left(\mu_{2}^{Rl},\nu_{2}^{Rl}\right)$	…	$\left(\mu_{i}^{Rl},\nu_{i}^{Rl}\right)$	…	$\left(\mu_{n}^{Rl},\nu_{n}^{Rl}\right)$
Flex	$\left(\mu_{1}^{Flex},\nu_{1}^{Flex}\right)$	$\left(\mu_{2}^{Flex},\nu_{2}^{Flex}\right)$	…	$\left(\mu_{i}^{Flex},\nu_{i}^{Flex}\right)$	…	$\left(\mu_{n}^{Flex},\nu_{n}^{Flex}\right)$
C	$\left(\mu_{1}^{C},\nu_{1}^{C}\right)$	$\left(\mu_{2}^{C},\nu_{2}^{C}\right)$	…	$\left(\mu_{i}^{C},\nu_{i}^{C}\right)$	…	$\left(\mu_{n}^{C},\nu_{n}^{C}\right)$

**Table 2 micromachines-12-00801-t002:** Basic information table of service node.

--Create table create table C3DS_NODES { ID INTERGER not null, NAME VARCHAR(100), //Order Name ORDERCATEGORY VARCHAR(50), //Order classification SERVICECATEGORY VARCHAR(50), //Service type PRINTMATERIAL NUMBER, //Printing material PROCESSINGTECHNOLOGY VARCHAR(50), //Processing technology STATUS VARCHAR(1), //Access status REMARK VARCHAR(200) }

**Table 3 micromachines-12-00801-t003:** The attribute table of C3DPS node data.

RowKey	TimeStamp	Columns			
		Orname	Orcategory	Prtechnology	Prmaterial
00001	0	Vatican gypsum relief	1	Gypsum 3D printing (PP)	Gypsum
00002	0	R2D2 robot	3	Selective laser sintering (SLS)	Metal powder
00003	1	Void cube model	1	Light curing (SLA)	Photosensitive polymer
00004	0	Eiffel Tower	7	Melt extrusion (FDM)	Thermoplastic material

**Table 4 micromachines-12-00801-t004:** The relation table of C3DPS node.

--Create table create table C3DS_NODES { ID NUMBER not null, SOURCE NUMBER, //Source node TARGET NUMBER, //Target node TYPE VARCHAR2(20), //Type WEIGHT NUMBER //Weight }

**Table 5 micromachines-12-00801-t005:** Candidates for 3D printing device resources.

Candidate Set	Atomic Service	Workshop Name	Equipment Model
$S_{1}$	$S_{11}$	Yourui 3D printing	HW-602
	$S_{12}$	Jiayi Hi-Tech	JOYE-4035
	$S_{13}$	Campus store	Aurora LVO A8
$S_{2}$	$S_{21}$	WINBO	WB-SH105
	$S_{22}$	Beien 3D	BANSOT M2
	$S_{23}$	3D printing workshop	Dimension SST 1200es
	$S_{24}$	The third brother of Hanbang	Corporate T1
$S_{3}$	$S_{31}$	Artful design workshop	SLM 500
	$S_{32}$	E-Plus-3D	EP-M100T
	$S_{33}$	Manheng	EOS-M290
$S_{4}$	$S_{41}$	Tongchuang 3D	MOONRAY
	$S_{42}$	Flashcast Technology Studio	Explorer
	$S_{43}$	Yourui 3D printing	DLP-1
$S_{5}$	$S_{51}$	Flashcast Technology Studio	Creator Pro
	$S_{52}$	Wuhan store	Second-generation 3D printing
	$S_{53}$	Jiayi Hi-Tech	JOYE-1010K
	$S_{54}$	Jiayi Hi-Tech	JOYE-1212E
$S_{6}$	$S_{61}$	Campus Station of College of Culture	FORTUS 200 mc
	$S_{62}$	3D printing workshop	ProJet 6000
	$S_{63}$	Yunle Design Studio	ULTRA
	$S_{64}$	High-precision printing	HOFTX2

## Data Availability

The data presented in this study are available on request from the first author.

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
