# Peer review of "Evaluation of Cloud 3D Printing Order Task Execution Based on the AHP-TOPSIS Optimal Set Algorithm and the Baldwin Effect"

_micromachines, 2021, doi:10.3390/mi12070801_

Round 1

Reviewer 1 Report

The authors present an interesting problem about cloud manufacturing in 3D_printing industry. However, the concrete example in 3D-printing cloud manufacturing should be further explained. For example,, why is 3D-printing benefit for cloud manufacturing? Especially, the services in Table 2 should further explained, including the name of the atomic services.

Author Response

Response to Reviewer 1 Comments

Dear Editors and Reviewers:    

Thank you for your letter and for the reviewers’ comments concerning our manuscript entitled “Evaluation of cloud 3D printing order task execution based on the AHP-TOPSIS optimal set algorithm and the Baldwin effect”. Those comments are all valuable and very helpful for revising and improving our paper, as well as the important guiding significance to our researches. We have studied comments carefully and have made correction which we hope meet with approval. Revised portion are marked in red in the paper.

 The main corrections in the paper and the responds to the reviewer’s comments are as following: 

Responds to the reviewer’s comments: 

Point 1:The authors present an interesting problem about cloud manufacturing in 3D_printing industry. However, the concrete example in 3D-printing cloud manufacturing should be further explained. For example, why is 3D-printing benefit for cloud manufacturing? Especially, the services in Table 2 should further explained, including the name of the atomic services.

Response 1: 

Comments:

(1) For example, why is 3D-printing benefit for cloud manufacturing? 

Response: On the background of the white-hot real market competition and the increasing popularity of public demand, a traditional model of large-scale production has been unable to meet the Innovation requirements in the "Internet +" Era. With the integration of Cloud Computing, Internet of things, Big Data, Mobile Internet and other advanced information technology incorporating into manufacturing technology deeply, service, network and intelligent has become the developing direction of the instrument. How to innovate the manufacturing service mode and achieve full sharing of manufacturing resources and cross-domain collaboration, and then support the majority of 3D printing industry to carry out a business of producer services, this problem has become one of the most important scientific proposition for the current advanced manufacturing fields. As a new exploration, cloud 3D printing service is a new method based on Ubiquitous Network, Data-driven, Shared services, Cross-border integration and Mass innovation, can support the people-centered management of distributed cloud 3D printing manufacturing resources and on-demand services based on Interconnected Personalized resources and Flexible service. With the deep integration of advanced manufacturing technologies and emerging network information technologies, many 3D printing companies have introduced cloud manufacturing models. In recent years, some emerging computer companies have built a series of 3D printing equipment sharing platforms based on the Internet, which has changed the previous dilemma that was restricted by professional design and professional processing technology and was unable to quickly turn customized ideas into actual products.

  • Especially, the services in Table 2 should further explained, including the name of the atomic services.

Response: We are very sorry for our negligence of Table 2. We have re-written this part according to the Reviewer’s suggestion.

Wheremeans Services=(Service 1(Point cloud data processing), Service 2(3D Reverse Engineering), Service 3(3D CAD modeling and design), Service 4(Slicing procedures for layered manufacturing technique), Service 5(simulation analysis), Service 6( 3D printing and post-processing)}. means three Candidates for 3D printing device resources.

Reviewer 2 Report

The manuscript presents a state of the art subject on the application of order task execution based on the AHP-TOPSIS. The manuscript is well structured however there are a series of considerations. 

The manuscript is out of scope for specific journal as it touches on a subject that is relevant to scheduling of 3D printing and not on the micro aspects of the process. It is more suitable for journals oriented in algorithms, manufacturing systems and cloud technologies.

The case study aspect of the manuscript needs to be strengthened.

Author Response

Response to Reviewer 2 Comments

Dear Editors and Reviewers:    

Thank you for your letter and for the reviewers’ comments concerning our manuscript entitled “Evaluation of cloud 3D printing order task execution based on the AHP-TOPSIS optimal set algorithm and the Baldwin effect”. Those comments are all valuable and very helpful for revising and improving our paper, as well as the important guiding significance to our researches. We have studied comments carefully and have made correction which we hope meet with approval. Revised portion are marked in red in the paper.

 The main corrections in the paper and the responds to the reviewer’s comments are as following: 

Responds to the reviewer’s comments: 

Point 2: The manuscript presents a state of the art subject on the application of order task execution based on the AHP-TOPSIS. The manuscript is well structured however there are a series of considerations. 

The manuscript is out of scope for specific journal as it touches on a subject that is relevant to scheduling of 3D printing and not on the micro aspects of the process. It is more suitable for journals oriented in algorithms, manufacturing systems and cloud technologies.

The case study aspect of the manuscript needs to be strengthened.

Response 2

Comments:

(1) The manuscript is out of scope for specific journal as it touches on a subject that is relevant to scheduling of 3D printing and not on the micro aspects of the process. It is more suitable for journals oriented in algorithms, manufacturing systems and cloud technologies.

Response: Thank you for your comments. The reason for submission of this paper is invited  from the publisher.

Dear Dr. Zhang,

We are organizing a Special Issue entitled “3D Printing Fabrication of Small Components” of the journal Micromachines (ISSN 2072-666X, https://www.mdpi.com/journal/micromachines), with Dr. Joan Josep Roa (Universitat Politècnica de Catalunya, Spain), Dr. Caroline Tardivat and Dr. Gemma Fargas serving as Guest Editors. Based on your relevant expertise, we cordially invite you to publish a *research* or *review* article with us and enjoy the benefits of open access publishing with Micromachines.

Dear Dr. Zhang,

Hope you are doing well. In September 2020, we invited you to contribute a paper to the Special Issue "3D Printing Fabrication of Small Components" of Micromachines (ISSN 2072-666X, https://www.mdpi.com/journal/micromachines). Guest Editors: Dr. Joan Josep Roa (Universitat Politcnica de Catalunya, Spain), Dr. Caroline Tardivat and Dr. Gemma Fargas Website: https://www.mdpi.com/journal/micromachines/special_issues/3D_Small_ Components

We would appreciate if you could let us know whether you are interested in contributing to this Special Issue?

Micromachines boasts a fast peer review process and online publishing of original results in open access format. Following peer review, a first decision is provided to authors within approximately 13.5 days of submission; acceptance to publication is undertaken in 1.9 days. An article processing charge of 1600 CHF (Swiss Francs) applies to each accepted paper (Please note that for papers submitted after 31 December 2020 an APC of 1800 CHF applies).

If you have any questions or require any additional information, please do not hesitate to contact me.

We hope that you will accept our invitation, and we look forward to hearing from you.

(2) The case study aspect of the manuscript needs to be strengthened.

Response: We are very sorry for our negligence of the case study aspect of the manuscript. The case study aspect of the manuscript has been strengthened.

5.1.  C3DPSs modeling

 Taking the construction of C3DPS network as an example, a computer programs is clustered to the structure of the C3DPS network. The relevant basic data of the network must be described as a graph in the form of a database table, and the database table is the structure of 3D printing service network. The simulation process of the C3DPS network is divided into the following steps:

Firstly, it express the information of C3DPSs. As the assembly resources of personalized design products have many types and different capabilities, the C3DPS template is expressed in XML format.

For example, some of the attributes of Line 3 of No.6 Studio of Urui 3D Printing include 3D printing service mode, 3D printing specifications and dimensions, printing accuracy, maximum printing speed, 3D printing equipment model, printing type, etc. The node relationship is shown in Figure 4 shown. As shown in Figure 5, it is a C3DPS expressed in XML format. At the same time, the C3DPS also uses XML format to express the logical relationship between resource-attribute-attribute value.

For example, some attributes of Line 3 of No.6 Studio of yourui 3D Printing include 3D printing service mode, 3D printing specifications and dimensions, printing accuracy, maximum printing speed, 3D printing equipment model, printing type, etc., and their node relationships are shown in Figure 4. As shown in Figure 5 and Figure 6, it is a C3DPS in XML format. At the same time, the C3DPS also uses XML format to express the logical relationship among resources, attributes and attribute values.

Figure 4The relationship diagram of C3DPS node.

Figure 5The description diagram of C3DPS.

Figure 6The Partial XML information graph of C3DPS template

Secondly, it determines the service node. Here, as the research object, 10 C3DPS providers is selected and established to the basic information table of the node, as shown in table 2.

Table 2. Basic information table of service node.

--Create table

create table C3DS_NODES {

ID INTERGER not null,

NAME VARCHAR(100),  //Order Name

ORDERCATEGORY VARCHAR(50),  //Order classification

SERVICECATEGORY VARCHAR(50),  //Service type

PRINTMATERIAL NUMBER,  //Printing material

PROCESSINGTECHNOLOGY VARCHAR(50),  //Processing technology

STATUS VARCHAR(1),  //Access status

REMARK VARCHAR(200)

}

Here, a program is incorporated in the software to cluster the structure of the C3DPS network, so as to describe the triple data of the cultural creative product personalized design product as a graph in the form of a database table, and generate the cultural creative product personalized Design the complex network topology of the product. Among them, the basic information of portrait 3D printing node and data attributes of C3DPS node are shown in Table 3.

Table 3. The attribute table of C3DPS node data. 

RowKey

TimeStamp

Columns

ORNAME

ORCATEGORY

PRTECHNOLOGY

PRMATERIAL

00001

0

Vatican gypsum relief

1

Gypsum 3D printing (PP)

Gypsum

00002

0

R2D2 robot

3

Selective laser sintering (SLS)

Metal powder

00003

1

Void cube model

1

Light curing (SLA)

Photosensitive polymer

00004

0

Eiffel Tower

7

Melt extrusion (FDM)

Thermoplastic material

Thirdly, it determines the alternative relationship between the candidate C3DPS providers (that is, there is a competitive relationship by the same or similar services providers), and create a node connection relationship table.

Because of the different comprehensive service capabilities of each C3DPS provider, it is necessary to investigate the each service node. For example, Jiayi Hi-Tech 3D scanner service provider provides 3D scanner outsourcing services, and the Worry 3D scanner Service providers also provide 3D scanner outsourcing services. The output service capabilities of both are the same, so their similar weight is 1. After the data of all C3DPSs are summarized and sorted, the node connection relationship table is generated according to the form of  database table, as shown in table 4.

Table 4. The relation table of C3DPS node. 

--Create table

create table C3DS_NODES

{

ID NUMBER not null,

SOURCE NUMBER,  //Source node

TARGET NUMBER,  //Target node

TYPE VARCHAR2(20), //Type

WEIGHT NUMBER   //Weight

}

Finally, the basic information table and connection relationship table of C3DPS node are imported into gephi-0.8.1 software for data analysis. As shown in Figure 7, the topology of C3DPS () network. It is composed of 33 nodes and 46 edges. Among them, the same color of nodes means that the types of C3DPSs are similar. The thickness of the edge represents the similarity between the two C3DPSs, that is, the service capability is replaceable.

Figure 7The network graph of C3DPS Network ()

Round 2

Reviewer 2 Report

The authors have improved the manuscript quality and content with the addition of the case study. In addition to the research presented, key pieces of literature are missing from the review.

Mourtzis, D., Vlachou, E., Milas, N., Tapoglou, N., & Mehnen, J. (2019). A cloud-based, knowledge-enriched framework for increasing machining efficiency based on machine tool monitoring. Proceedings of the Institution of Mechanical Engineers, Part B: Journal of Engineering Manufacture233(1), 278-292.

Tapoglou, N., Mehnen, J., Vlachou, A., Doukas, M., Milas, N., & Mourtzis, D. (2015). Cloud-based platform for optimal machining parameter selection based on function blocks and real-time monitoring. Journal of Manufacturing Science and Engineering137(4).

Li, W., & Mehnen, J. (2013). Cloud manufacturing. Distributed Computing Technologies for Global and Sustainable Manufacturing, Springer Series in Advanced Manufacturing. Google Scholar Google Scholar Digital Library Digital Library.

This manuscript is a resubmission of an earlier submission. The following is a list of the peer review reports and author responses from that submission.